# Constrained convex minimization
# via model-based excessive gap

**Quoc Tran-Dinh** and **Volkan Cevher**
Laboratory for Information and Inference Systems (LIONS)
École Polytechnique Fédérale de Lausanne (EPFL), CH1015-Lausanne, Switzerland
{quoc.trandinh, volkan.cevher}@epfl.ch

## Abstract

We introduce a model-based excessive gap technique to analyze first-order primal-dual methods for constrained convex minimization. As a result, we construct first-order primal-dual methods with optimal convergence rates on the primal objective residual and the primal feasibility gap of their iterates separately. Through a dual smoothing and prox-center selection strategy, our framework subsumes the augmented Lagrangian, alternating direction, and dual fast-gradient methods as special cases, where our rates apply.

## 1 Introduction

In [1], Nesterov introduced a primal-dual technique, called the *excessive gap*, for constructing and analyzing first-order methods for nonsmooth and unconstrained convex optimization problems. This paper builds upon the same idea for constructing and analyzing algorithms for the following a class of *constrained* convex problems, which captures a surprisingly broad set of applications [2, 3, 4, 5]:

$$f^\star := \min_{\mathbf{x} \in \mathbb{R}^n} \{f(\mathbf{x}) : \mathbf{A}\mathbf{x} = \mathbf{b}, \ \mathbf{x} \in \mathcal{X}\}, \tag{1}$$

where $f : \mathbb{R}^n \to \mathbb{R} \cup \{+\infty\}$ is a proper, closed and convex function; $\mathcal{X} \subseteq \mathbb{R}^n$ is a nonempty, closed and convex set; and $\mathbf{A} \in \mathbb{R}^{m \times n}$ and $\mathbf{b} \in \mathbb{R}^m$ are given.

In the sequel, we show how Nesterov's excessive gap relates to the smoothed gap function for a variational inequality that characterizes the optimality condition of (1). In the light of this connection, we enforce a simple linear model on the excessive gap, and use it to develop efficient first-order methods to numerically approximate an optimal solution $\mathbf{x}^\star$ of (1). Then, we rigorously characterize how the following structural assumptions on (1) affect their computational efficiency:

**Structure 1: Decomposability.** We say that problem (1) is *p-decomposable* if its objective function $f$ and its feasible set $\mathcal{X}$ can be represented as follows:

$$f(\mathbf{x}) := \sum_{i=1}^{p} f_i(\mathbf{x}_i), \text{ and } \mathcal{X} := \prod_{i=1}^{p} \mathcal{X}_i, \tag{2}$$

where $\mathbf{x}_i \in \mathbb{R}^{n_i}$, $\mathcal{X}_i \in \mathbb{R}^{n_i}$, $f_i : \mathbb{R}^{n_i} \to \mathbb{R} \cup \{+\infty\}$ is proper, closed and convex for $i = 1, \ldots, p$, and $\sum_{i=1}^{p} n_i = n$. Decomposability naturally arises in machine learning applications such as group sparsity linear recovery, consensus optimization, and duality of empirical risk minimization problems [5]. As an important example, a composite convex minimization problem $\min_{\mathbf{x}_1}\{f_1(\mathbf{x}_1) + f_2(\mathbf{K}\mathbf{x}_1)\}$ can be cast into (1) with a 2-decomposable structure using an intermediate variable $\mathbf{x}_2 = \mathbf{K}\mathbf{x}_1$ to represent the linear constraints. Decomposable structure immediately supports parallel and distributed implementations in synchronous hardware architectures.

**Structure 2: Proximal tractability.** By proximal tractability, we mean that the computation of the following operation with a given proper, closed and convex function $g$ is "efficient" (e.g., by a closed form solution or by polynomial time algorithms) [6]:

$$\mathrm{prox}_g(\mathbf{z}) := \arg\min_{\mathbf{w} \in \mathbb{R}^{n_z}} \{g(\mathbf{w}) + (1/2)\|\mathbf{w} - \mathbf{z}\|^2\}. \tag{3}$$

When the constraint $\mathbf{z} \in \mathcal{Z}$ is available, we consider the proximal operator of $g(\cdot) + \delta_{\mathcal{Z}}(\cdot)$ instead of $g$, where $\delta_{\mathcal{Z}}$ is the indicator function of $\mathcal{Z}$. Many smooth and non-smooth functions have tractable proximal operators such as norms, and the projection onto a simple set [3, 7, 4, 5].

**Scalable algorithms for constrained convex minimization and their limitations.**
We can obtain scalable numerical solutions of (1) when we augment the objective $f$ with simple penalty functions on the constraints. Despite the fundamental difficulties in choosing the penalty parameter, this approach enhances our computational capabilities as well as numerical robustness since we can apply modern proximal gradient, alternating direction, and primal-dual methods. Unfortunately, existing approaches invariably feature one or both of the following two limitations:

**Limitation 1: Non-ideal convergence characterizations.** Ideally, the convergence rate characterization of a first-order algorithm for solving (1) must simultaneously establish for its iterates $\mathbf{x}^k \in \mathcal{X}$ both on the objective residual $f(\mathbf{x}^k) - f^\star$ and on the primal feasibility gap $\|\mathbf{A}\mathbf{x}^k - \mathbf{b}\|$ of its linear constraints. The constraint feasibility is critical so that the primal convergence rate has any significance. Rates on a joint of the objective residual and feasibility gap is not necessarily meaningful since (1) is a constrained problem and $f(\mathbf{x}^k) - f^\star$ can easily be negative at all times as compared to the unconstrained setting, where we trivially have $f(\mathbf{x}^k) - f^\star \geq 0$.

Hitherto, the convergence results of state-of-the-art methods are far from ideal; see Table 1 in [28]. Most algorithms have guarantees in the ergodic sense [8, 9, 10, 11, 12, 13, 14] with non-optimal rates, which diminishes the practical performance; they rely on special function properties to improve convergence rates on the function and feasibility [12, 15], which reduces the scope of their applicability; they provide rates on dual functions [16], or a weighted primal residual and feasibility score [13], which does not necessarily imply convergence on the primal residual or the feasibility; or they obtain convergence rate on the gap function value sequence composed both the primal and dual variables via variational inequality and gap function characterizations [8, 10, 11], where the rate is scaled by a diameter parameter of the dual feasible set which is not necessary bounded.

**Limitation 2: Computational inflexibility.** Recent theoretical developments customize algorithms to special function classes for scalability, such as convex functions with global Lipschitz gradient and strong convexity. Unfortunately, these algorithms often require knowledge of function class parameters (e.g., the Lipschitz constant and the strong convexity parameter); they do not address the full scope of (1) (e.g., with self-concordant [barrier] functions or fully non-smooth decompositions); and they often have complicated algorithmic implementations with backtracking steps, which can create computational bottlenecks. These issues are compounded by their penalty parameter selection, which can significantly decrease numerical efficiency [17]. Moreover, they lack a natural ability to handle $p$-decomposability in a parallel fashion at optimal rates.

**Our specific contributions**

To this end, this paper addresses the question: "Is it possible to efficiently solve (1) using only the proximal tractability assumption with rigorous global convergence rates on the objective residual and the primal feasibility gap?" The answer is indeed positive provided that there exists a solution in a bounded feasible set $\mathcal{X}$. Surprisingly, we can still leverage favorable function classes for fast convergence, such as strongly convex functions, and exploit $p$-decomposability at optimal rates.

Our characterization is radically different from existing results, such as in [18, 8, 19, 9, 10, 11, 12, 13]. Specifically, we unify primal-dual methods [20, 21], smoothing (both for Bregman distances and for augmented Lagrangian functions) [22, 21], and the excessive gap function technique [1] in one. As a result, we develop an efficient algorithmic framework for solving (1), which covers augmented Lagrangian method [23, 24], [preconditioned] alternating direction method-of-multipliers ([P]ADMM) [8] and fast dual descent methods [18] as special cases.

Based on the new technique, we establish rigorous convergence rates for a few well-known primal-dual methods, which is optimal (in the sense of first order black-box models [25]) given our particular assumptions. We also discuss adaptive strategies for trading-off between the objective residual $|f(\mathbf{x}^k) - f^\star|$ and the feasibility gap $\|\mathbf{A}\mathbf{x}^k - \mathbf{b}\|$, which enhance practical performance. Finally, we describe how strong convexity of $f$ can be exploited, and numerically illustrate theoretical results.

## 2 Preliminaries

**2.1. A semi-Bregman distance.** Let $\mathcal{Z}$ be a nonempty, closed convex set in $\mathbb{R}^{n_z}$. A nonnegative, continuous and $\mu_b$-strongly convex function $b$ is called a $\mu_b$-proximity function or prox-function of $\mathcal{Z}$ if $\mathcal{Z} \subseteq \mathrm{dom}(b)$. Then $\mathbf{z}_c := \mathrm{argmin}_{\mathbf{z} \in \mathcal{Z}} b(\mathbf{z})$ exists and is unique, called the center point of $b$. Given a smooth $\mu_b$-prox-function $b$ of $\mathcal{Z}$ (with $\mu_b = 1$), we define $d_b(\mathbf{z}, \hat{\mathbf{z}}) := b(\hat{\mathbf{z}}) - b(\mathbf{z}) - \nabla b(\mathbf{z})^T(\hat{\mathbf{z}} - \mathbf{z})$, $\forall \mathbf{z}, \hat{\mathbf{z}} \in \mathrm{dom}(b)$, as the Bregman distance between $\mathbf{z}$ and $\hat{\mathbf{z}}$ given $b$. As an example, with $b(\mathbf{z}) := (1/2)\|\mathbf{z}\|_2^2$, we have $d_b(\mathbf{z}, \hat{\mathbf{z}}) = (1/2)\|\mathbf{z} - \hat{\mathbf{z}}\|_2^2$, which is the Euclidean distance.

In order to unify both the Bregman distance and augmented Lagrangian smoothing methods, we introduce a new semi-Bregman distance $d_b(\mathbf{Sx}, \mathbf{Sx}_c)$ between $\mathbf{x}$ and $\mathbf{x}_c$, given matrix $\mathbf{S}$. Since $\mathbf{S}$ is not necessary square, we use the prefix "semi" for this measure. We also denote by:

$$D_{\mathcal{X}}^{\mathbf{S}} := \sup\{d_b(\mathbf{Sx}, \mathbf{Sx}_c) : \mathbf{x}, \mathbf{x}_c \in \mathcal{X}\}, \tag{4}$$

the semi-diameter of $\mathcal{X}$. If $\mathcal{X}$ is bounded, then $0 \leq D_{\mathcal{X}}^{\mathbf{S}} < +\infty$.

**2.2. The dual problem of** (1). Let $\mathcal{L}(\mathbf{x}, \mathbf{y}) := f(\mathbf{x}) + \mathbf{y}^T(\mathbf{Ax} - \mathbf{b})$ be the Lagrange function of (1), where $\mathbf{y} \in \mathbb{R}^m$ is the Lagrange multipliers. The dual problem of (1) is defined as:

$$g^\star := \max_{\mathbf{y} \in \mathbb{R}^m} g(\mathbf{y}), \tag{5}$$

where $g$ is the dual function, which is defined as:

$$g(\mathbf{y}) := \min_{\mathbf{x} \in \mathcal{X}}\{f(\mathbf{x}) + \mathbf{y}^T(\mathbf{Ax} - \mathbf{b})\}. \tag{6}$$

Let us denote by $\mathbf{x}^\star(\mathbf{y})$ the solution of (6) for a given $\mathbf{y} \in \mathbb{R}^m$. Corresponding to $\mathbf{x}^\star(\mathbf{y})$, we also define the domain of $g$ as $\mathrm{dom}(g) := \{\mathbf{y} \in \mathbb{R}^m : \mathbf{x}^\star(\mathbf{y}) \text{ exists}\}$. If $f$ is continuous on $\mathcal{X}$ and if $\mathcal{X}$ is bounded, then $\mathbf{x}^\star(\mathbf{y})$ exists for all $\mathbf{y} \in \mathbb{R}^m$. Unfortunately, $g$ is nonsmooth, and numerical solutions of (5) are difficult [25]. In general, we have $g(\mathbf{y}) \leq f(\mathbf{x})$ which is the weak-duality condition in convex analysis. To guarantee strong duality, i.e., $f^\star = g^\star$ for (1) and (5), we need an assumption:

**Assumption A. 1.** *The solution set $\mathcal{X}^\star$ of* (1) *is nonempty. The function $f$ is proper, closed and convex. In addition, either $\mathcal{X}$ is a polytope or the Slater condition holds, i.e.: $\{\mathbf{x} \in \mathbb{R}^n : \mathbf{Ax} = \mathbf{b}\} \cap \mathrm{relint}(\mathcal{X}) \neq \emptyset$, where $\mathrm{relint}(\mathcal{X})$ is the relative interior of $\mathcal{X}$.*

Under Assumption A.1, the solution set $\mathcal{Y}^\star$ of (5) is also nonempty and bounded. Moreover, the strong duality holds, i.e., $f^\star = g^\star$. Any point $(\mathbf{x}^\star, \mathbf{y}^\star) \in \mathcal{X}^\star \times \mathcal{Y}^\star$ is a primal-dual solution to (1) and (5), and is also a saddle point of $\mathcal{L}$, i.e., $\mathcal{L}(\mathbf{x}^\star, \mathbf{y}) \leq \mathcal{L}(\mathbf{x}^\star, \mathbf{y}^\star) \leq \mathcal{L}(\mathbf{x}, \mathbf{y}^\star), \forall(\mathbf{x}, \mathbf{y}) \in \mathcal{X} \times \mathbb{R}^m$.

**2.3. Mixed-variational inequality formulation and the smoothed gap function.** We use $\mathbf{w} := [\mathbf{x}; \mathbf{y}] \in \mathbb{R}^n \times \mathbb{R}^m$ to denote the primal-dual variable, and $F(\mathbf{w}) := \begin{bmatrix} \mathbf{A}^T\mathbf{y} \\ \mathbf{b} - \mathbf{Ax} \end{bmatrix}$ to denote a partial Karush-Kuhn-Tucker (KKT) mapping. Then, we can write the optimality condition of (1) as:

$$f(\mathbf{x}) - f(\mathbf{x}^\star) + F(\mathbf{w}^\star)^T(\mathbf{w} - \mathbf{w}^\star) \geq 0, \quad \forall \mathbf{w} \in \mathcal{X} \times \mathbb{R}^m, \tag{7}$$

which is known as the *mixed-variational inequality* (MVIP) [26]. If we define $\mathcal{W} := \mathcal{X} \times \mathbb{R}^m$ and:

$$G(\mathbf{w}^\star) := \max_{\mathbf{w} \in \mathcal{W}}\left\{f(\mathbf{x}^\star) - f(\mathbf{x}) + F(\mathbf{w}^\star)^T(\mathbf{w}^\star - \mathbf{w})\right\}, \tag{8}$$

then $G$ is known as the Auslender gap function of (7) [27]. By the definition of $F$, we can see that:

$$G(\mathbf{w}^\star) := \max_{(\mathbf{x}, \mathbf{y}) \in \mathcal{W}}\left\{f(\mathbf{x}^\star) - f(\mathbf{x}) - (\mathbf{Ax} - \mathbf{b})^T\mathbf{y}^\star\right\} = f(\mathbf{x}^\star) - g(\mathbf{y}^\star) \geq 0.$$

It is clear that $G(\mathbf{w}^\star) = 0$ if and only if $\mathbf{w}^\star := [\mathbf{x}^\star; \mathbf{y}^\star] \in \mathcal{W}^\star := \mathcal{X}^\star \times \mathcal{Y}^\star$—i.e., the strong duality.

Since $G$ is generally nonsmooth, we strictly smooth it by adding an augmented convex function:

$$d_{\gamma\beta}(\mathbf{w}) \equiv d_{\gamma\beta}(\mathbf{x}, \mathbf{y}) := \gamma d_b(\mathbf{Sx}, \mathbf{Sx}_c) + (\beta/2)\|\mathbf{y}\|^2, \tag{9}$$

where $d_b$ is a Bregman distance, $\mathbf{S}$ is a given matrix, and $\gamma, \beta > 0$ are *smoothness parameters*. The smoothed gap function for $G$ is defined as:

$$G_{\gamma\beta}(\bar{\mathbf{w}}) := \max_{\mathbf{w} \in \mathcal{W}}\left\{f(\bar{\mathbf{x}}) - f(\mathbf{x}) + F(\bar{\mathbf{w}})^T(\bar{\mathbf{w}} - \mathbf{w}) - d_{\gamma\beta}(\mathbf{w})\right\}, \tag{10}$$

where $F$ is defined in (7). The function $G_{\gamma\beta}$ can be considered as smoothed gap function for the MVIP (7). By the definition of $G$ and $G_{\gamma\beta}$, we can easily show that:

$$G_{\gamma\beta}(\bar{\mathbf{w}}) \leq G(\bar{\mathbf{w}}) \leq G_{\gamma\beta}(\bar{\mathbf{w}}) + \max\{d_{\gamma\beta}(\mathbf{w}) : \mathbf{w} \in \mathcal{W}\}, \tag{11}$$

which is key to develop the algorithm in the next section.

Problem (10) is convex, and its solution $\mathbf{w}_{\gamma\beta}^\star(\bar{\mathbf{w}})$ can be computed as:

$$\mathbf{w}_{\gamma\beta}^\star(\bar{\mathbf{w}}) := [\mathbf{x}_\gamma^\star(\bar{\mathbf{y}}); \mathbf{y}_\beta^\star(\bar{\mathbf{x}})] \Leftrightarrow \begin{cases} \mathbf{x}_\gamma^\star(\bar{\mathbf{y}}) := \underset{\mathbf{x} \in \mathcal{X}}{\mathrm{argmin}}\left\{f(\mathbf{x}) + \mathbf{y}^T(\mathbf{Ax} - \mathbf{b}) + \gamma d_b(\mathbf{Sx}, \mathbf{Sx}_c)\right\} \\ \mathbf{y}_\beta^\star(\bar{\mathbf{x}}) := \beta^{-1}(\mathbf{A}\bar{\mathbf{x}} - \mathbf{b}). \end{cases} \tag{12}$$

In this case, the following concave function:

$$g_\gamma(\mathbf{y}) := \min_{\mathbf{x} \in \mathcal{X}}\left\{f(\mathbf{x}) + \mathbf{y}^T(\mathbf{Ax} - \mathbf{b}) + \gamma d_b(\mathbf{Sx}, \mathbf{Sx}_c)\right\}, \tag{13}$$

can be considered as a smooth approximation of the dual function $g$ defined by (6).

**2.4. Bregman distance smoother vs. augmented Lagrangian smoother.** Depending on the choice of $\mathbf{S}$ and $\mathbf{x}_c$, we deal with two smoothers as follows:

1. If we choose $\mathbf{S} = \mathbb{I}$, the identity matrix, and $\mathbf{x}_c$ is then center point of $b$, then we obtain a Bregman distance smoother.

2. If we choose $\mathbf{S} = \mathbf{A}$, and $\mathbf{x}_c \in \mathcal{X}$ such that $\mathbf{A}\mathbf{x}_c = \mathbf{b}$, then we have the augmented Lagrangian smoother.

Clearly, with both smoothing techniques, the function $g_\gamma$ is smooth and concave. Its gradient is Lipschitz continuous with the Lipschitz constant $L_\gamma^g := \gamma^{-1}\|\mathbf{A}\|^2$ and $L_\gamma^g := \gamma^{-1}$, respectively.

# 3 Construction and analysis of a class of first-order primal-dual algorithms

**3.1. Model-based excessive gap technique for** (1). Since $G(\mathbf{w}^\star) = 0$ iff $\mathbf{w}^\star = [\mathbf{x}^\star; \mathbf{y}^\star]$ is a primal-dual optimal solution of (1)-(5). The goal is to construct a sequence $\{\bar{\mathbf{w}}^k\}$ such that $G(\bar{\mathbf{w}}^k) \to 0$, which implies that $\{\bar{\mathbf{w}}^k\}$ converges to $\mathbf{w}^\star$. As suggested by (11), if we can construct two sequences $\{\bar{\mathbf{w}}^k\}$ and $\{(\gamma_k, \beta_k)\}$ such that $G_{\gamma_k\beta_k}(\bar{\mathbf{w}}^k) \to 0^+$ as $\gamma_k\beta_k \downarrow 0^+$, then $G(\bar{\mathbf{w}}^k) \to 0$.

Inspired by Nesterov's excessive gap idea in [1], we construct the following model-based excessive gap condition for (1) in order to achieve our goal.

**Definition 1** (Model-based Excessive Gap). *Given $\bar{\mathbf{w}}^k \in \mathcal{W}$ and $(\gamma_k, \beta_k) > 0$, a new point $\bar{\mathbf{w}}^{k+1} \in \mathcal{W}$ and $(\gamma_{k+1}, \beta_{k+1}) > 0$ so that $\gamma_{k+1}\beta_{k+1} < \gamma_k\beta_k$ is said to be firmly contractive (w.r.t. $G_{\gamma\beta}$ defined by (10)) when it holds for $G_{\gamma_k\beta_k}$ that:*

$$G_{k+1}(\bar{\mathbf{w}}^{k+1}) \leq (1 - \tau_k)G_k(\bar{\mathbf{w}}^k) - \psi_k, \tag{14}$$

*where $G_k := G_{\gamma_k\beta_k}$, $\tau_k \in [0, 1)$ and $\psi_k \geq 0$.*

From Definition 1, if $\{\bar{\mathbf{w}}^k\}$ and $\{(\gamma_k, \beta_k)\}$ satisfy (14), then we have $G_k(\bar{\mathbf{w}}^k) \leq \omega_k G_0(\bar{\mathbf{w}}^0) - \Psi_k$ by induction, where $\omega_k := \prod_{j=0}^{k-1}(1 - \tau_j)$ and $\Psi_k := \psi_0 + \sum_{j=1}^{k-1}\prod_{i=0}^{j-1}(1 - \tau_i)\psi_j$. If $G_0(\bar{\mathbf{w}}^0) \leq 0$, then we can bound the objective residual $|f(\bar{\mathbf{x}}^k) - f^\star|$ and the primal feasibility $\|\mathbf{A}\bar{\mathbf{x}}^k - \mathbf{b}\|$ of (1):

**Lemma 1** ([28]). *Let $G_{\gamma\beta}$ be defined by (10). Let $\{\bar{\mathbf{w}}^k\}_{k\geq 0} \subset \mathcal{W}$ and $\{(\gamma_k, \beta_k)\}_{k\geq 0} \in \mathbb{R}_{++}^2$ be the sequences that satisfy (14). Then, it holds that:*

$$\begin{cases} -\big[2\beta_k D_\mathcal{Y}^\star + (2\gamma_k\beta_k D_\mathcal{X}^\mathbf{S})^{1/2}\big]D_\mathcal{Y}^\star \leq & f(\bar{\mathbf{x}}^k) - f^\star \leq \gamma_k D_\mathcal{X}^\mathbf{S}, \\ & \|\mathbf{A}\bar{\mathbf{x}}^k - \mathbf{b}\| \leq 2\beta_k D_\mathcal{Y}^\star + (2\gamma_k\beta_k D_\mathcal{X}^\mathbf{S})^{1/2}, \end{cases} \tag{15}$$

*where $D_\mathcal{Y}^\star := \min\{\|\mathbf{y}^\star\|_2 : \mathbf{y}^\star \in \mathcal{Y}^\star\}$, which is the norm of a minimum norm dual solutions.*

Hence, we can derive algorithms based $(\gamma_k, \beta_k)$ with a predictable convergence rate via (15). In the sequel, we manipulate $\tau_k$ and $\psi_k$ to do just that in order to preserve (14) á la Nesterov [1]. Finally, we say that $\bar{\mathbf{x}}^k \in \mathcal{X}$ is an $\varepsilon$-solution of (1) if $|f(\bar{\mathbf{x}}^k) - f^*| \leq \varepsilon$ and $\|\mathbf{A}\bar{\mathbf{x}}^k - \mathbf{b}\| \leq \varepsilon$.

**3.2. Initial points.** We first show how to compute an initial point $\mathbf{w}^0$ such that $G_0(\bar{\mathbf{w}}^0) \leq 0$.

**Lemma 2** ([28]). *Given $\mathbf{x}_c^0 \in \mathcal{X}$, $\bar{\mathbf{w}}^0 := [\bar{\mathbf{x}}^0; \bar{\mathbf{y}}^0] \in \mathcal{W}$ is computed by:*

$$\begin{cases} \bar{\mathbf{x}}^0 & = \mathbf{x}_{\gamma_0}^*(0^m) := \arg\min_{\mathbf{x}\in\mathcal{X}}\big\{f(\mathbf{x}) + (\gamma_0/2)d_b(\mathbf{Sx}, \mathbf{Sx}_c^0)\big\}, \\ \bar{\mathbf{y}}^0 & = \mathbf{y}_{\beta_0}^*(\bar{\mathbf{x}}^0) := \beta_0^{-1}(\mathbf{A}\bar{\mathbf{x}}^0 - \mathbf{b}). \end{cases} \tag{16}$$

*satisfies $G_{\gamma_0\beta_0}(\bar{\mathbf{w}}^0) \leq -\gamma_0 d_p(\mathbf{S}\bar{\mathbf{x}}^0, \mathbf{Sx}_c) \leq 0$ provided that $\beta_0\gamma_0 \geq \bar{L}^g$, where $\bar{L}^g$ is the Lipschitz constant of $\nabla g_\gamma$ with $g_\gamma$ given Subsection 2.4.*

**3.3. An algorithmic template.** Algorithm 1 combines the above ingredients for solving (1). We observe that the key computational step of Algorithm 1 is Step 3, where we update $[\bar{\mathbf{x}}^{k+1}; \bar{\mathbf{y}}^{k+1}]$. In the algorithm, we provide two update schemes (1P2D) and (2P1D) based on the updates of the primal or dual variables. The primal step $\mathbf{x}_{\gamma_k}^*(\bar{\mathbf{y}}^k)$ is calculated via (12). At line 3 of (2P1D), the operator $\text{prox}_{\beta f}^\mathbf{S}$ is computed as:

$$\text{prox}_{\beta f}^\mathbf{S}(\hat{\mathbf{x}}, \hat{\mathbf{y}}) := \arg\min_{\mathbf{x}\in\mathcal{X}}\big\{f(\mathbf{x}) + \hat{\mathbf{y}}^T\mathbf{A}(\mathbf{x} - \hat{\mathbf{x}}) + \beta^{-1}d_b(\mathbf{Sx}, \mathbf{S}\hat{\mathbf{x}})\big\}, \tag{17}$$

where we overload the notation of the proximal operator $\text{prox}$ defined above. At Step 2 of Algorithm 1, if we choose $\mathbf{S} := \mathbb{I}$, i.e., $d_b(\mathbf{Sx}, \mathbf{Sx}_c) := d_b(\mathbf{x}, \mathbf{x}_c)$ for $\mathbf{x}_c$ being the center point of $b$, then we set $\bar{L}^g := \|\mathbf{A}\|^2$. If $\mathbf{S} := \mathbf{A}$, i.e., $d_b(\mathbf{Sx}, \mathbf{Sx}_c) := (1/2)\|\mathbf{Ax} - \mathbf{b}\|^2$, then we set $\bar{L}^g := 1$.

Theorem 1 characterizes three variants of Algorithm 1, whose proof can be found in [28].

**Algorithm 1:** (*A primal-dual algorithmic template using model-based excessive gap*)

---

**Inputs:** Fix $\gamma_0 > 0$. Choose $c_0 \in (-1, 1]$.
**Initialization:**
1: Compute $a_0 := 0.5(1 + c_0 + \sqrt{4(1 - c_0) + (1 + c_0)^2})$, $\tau_0 := a_0^{-1}$, and $\beta_0 := \gamma_0^{-1} \bar{L}^g$ (c.f. the text).
2: Compute $[\bar{\mathbf{x}}^0; \bar{\mathbf{y}}^0]$ as (16) in Lemma 2.
**For** $k = 0$ **to** $k_{\max}$**, perform:**
3: If **stopping_criterion**, terminate. Otherwise, use one of the following update schemes:

$$
(\text{2P1D}) : \begin{cases} \hat{\mathbf{x}}^k := (1 - \tau_k)\bar{\mathbf{x}}^k + \tau_k \mathbf{x}_{\gamma_k}^*(\bar{\mathbf{y}}^k) \\ \hat{\mathbf{y}}^k := \beta_{k+1}^{-1}(\mathbf{A}\hat{\mathbf{x}}^k - \mathbf{b}) \\ \bar{\mathbf{x}}^{k+1} := \text{prox}_{\beta_{k+1}f}^{\mathbf{S}}(\hat{\mathbf{x}}^k, \hat{\mathbf{y}}^k) \\ \bar{\mathbf{y}}^{k+1} := (1 - \tau_k)\bar{\mathbf{y}}^k + \tau_k \hat{\mathbf{y}}^k. \end{cases} \quad (\text{1P2D}) : \begin{cases} \bar{\mathbf{y}}_k^\star := \beta_k^{-1}(\mathbf{A}\bar{\mathbf{x}}^k - \mathbf{b}), \\ \hat{\mathbf{y}}^k := (1 - \tau_k)\bar{\mathbf{y}}^k + \tau_k \bar{\mathbf{y}}_k^\star, \\ \bar{\mathbf{x}}^{k+1} := (1 - \tau_k)\bar{\mathbf{x}}^k + \tau_k \mathbf{x}_{\gamma_k}^*(\hat{\mathbf{y}}^k), \\ \bar{\mathbf{y}}^{k+1} := \hat{\mathbf{y}}^k + \gamma_k (\mathbf{A}\mathbf{x}_{\gamma_k}^*(\hat{\mathbf{y}}^k) - \mathbf{b}). \end{cases}
$$

4: Update $\beta_{k+1} := (1 - \tau_k)\beta_k$ and $\gamma_{k+1} := (1 - c_k \tau_k)\gamma_k$. Update $c_{k+1}$ from $c_k$ (optional).
5: Update $a_{k+1} := 0.5(1 + c_{k+1} + \sqrt{4a_k^2 + (1 - c_{k+1})^2})$ and set $\tau_{k+1} := a_{k+1}^{-1}$.
**End For**

---

**Theorem 1.** *Let $\{(\bar{\mathbf{x}}^k, \bar{\mathbf{y}}^k)\}$ be the sequence generated by Algorithm 1 after $k$ iterations. Then:*

*If $\mathbf{S} = \mathbf{A}$, i.e., using the augmented Lagrangian smoother, $\gamma_0 := \sqrt{\bar{L}^g} = 1$, and $c_k := 0$, then the* (1P2D) *update satisfies:*

$$
\begin{cases} \|\mathbf{A}\bar{\mathbf{x}}^k - \mathbf{b}\|_2 \leq \frac{8D_{\mathcal{Y}}^\star}{(k+1)^2}, \\ -\frac{1}{2}\|\mathbf{A}\bar{\mathbf{x}}^k - \mathbf{b}\|_2^2 - D_{\mathcal{Y}}^\star \|\mathbf{A}\bar{\mathbf{x}}^k - \mathbf{b}\|_2 \leq f(\bar{\mathbf{x}}^k) - f^\star \leq 0, \end{cases} \quad (18)
$$

*for all $k \geq 0$. As a consequence, the worst-case analytical complexity of Algorithm 1 to achieve an $\varepsilon$-solution $\bar{\mathbf{x}}^k$ is $\mathcal{O}(\sqrt{\varepsilon})$.*

*If $\mathbf{S} = \mathbb{I}$, i.e., using the Bregman distance smoother, $\gamma_0 := \sqrt{\bar{L}^g} = \|\mathbf{A}\|$, and $c_k := 1$, then, for the* (2P1D) *scheme, we have:*

$$
(\text{2P1D}) : \begin{cases} \|\mathbf{A}\bar{\mathbf{x}}^k - \mathbf{b}\| \leq \frac{\|\mathbf{A}\|(2D_{\mathcal{Y}}^\star + \sqrt{2D_{\mathcal{X}}^\mathbb{I}})}{k+1}, \\ -D_{\mathcal{Y}}^\star \|\mathbf{A}\bar{\mathbf{x}}^k - \mathbf{b}\| \leq f(\bar{\mathbf{x}}^k) - f^\star \leq \frac{\|\mathbf{A}\|}{k+1} D_{\mathcal{X}}^\mathbb{I}. \end{cases} \quad (19)
$$

*Similarly, if $\gamma_0 := \frac{2\sqrt{2}\|\mathbf{A}\|}{K+1}$ and $c_k := 0$ for all $k = 0, 1, \ldots, K$, then, for the* (1P2D) *scheme, we have:*

$$
(\text{1P2D}) : \begin{cases} \|\mathbf{A}\bar{\mathbf{x}}^K - \mathbf{b}\| \leq \frac{2\sqrt{2}\|\mathbf{A}\|(D_{\mathcal{Y}}^\star + \sqrt{D_{\mathcal{X}}^\mathbb{I}})}{(K+1)}, \\ -D_{\mathcal{Y}}^\star \|\mathbf{A}\bar{\mathbf{x}}^K - \mathbf{b}\| \leq f(\bar{\mathbf{x}}^K) - f^\star \leq \frac{2\sqrt{2}\|\mathbf{A}\|}{(K+1)} D_{\mathcal{X}}^\mathbb{I}. \end{cases} \quad (20)
$$

*Hence, the worst-case analytical complexity to achieve an $\varepsilon$-solution $\bar{\mathbf{x}}^k$ of (1) is $\mathcal{O}(\varepsilon^{-1})$.*

The (1P2D) scheme has close relationship to some well-known primal dual methods we describe below. Unfortunately, 1P2D has the drawback of fixing the total number of iterations *a priori*, which 2P1D can avoid at the expense of one more proximal operator calculation at each iteration.

**3.4. Impact of strong convexity.** We can improve the above schemes when $f \in \mathcal{F}_\mu$, i.e., $f$ is strongly convex with parameter $\mu_f > 0$. The dual function $g$ given in (6) is smooth and Lipschitz gradient with $L_f^g := \mu_f^{-1}\|\mathbf{A}\|^2$. Let us illustrate this when $\mathbf{S} = \mathbb{I}$ and using the (1P2D) scheme as:

$$
(\text{1P2D}_\mu) \begin{cases} \hat{\mathbf{y}}^k := (1 - \tau_k)\bar{\mathbf{y}}^k + \tau_k \mathbf{y}_{\beta_k}^\star(\bar{\mathbf{x}}^k), \\ \bar{\mathbf{x}}^{k+1} := (1 - \tau_k)\bar{\mathbf{x}}^k + \tau_k \mathbf{x}^\star(\hat{\mathbf{y}}^k), \\ \bar{\mathbf{y}}^{k+1} := \hat{\mathbf{y}}^k + \frac{1}{L_f^g}(\mathbf{A}\mathbf{x}^\star(\hat{\mathbf{y}}^k) - \mathbf{b}). \end{cases}
$$

We can still choose the starting point as in (16) with $\beta_0 := L_f^g$. The parameters $\beta_k$ and $\tau_k$ at Steps 4 and 5 of Algorithm 1 are updated as $\beta_{k+1} := (1 - \tau_k)\beta_k$, and $\tau_{k+1} := \frac{\tau_k}{2}(\sqrt{\tau_k^2 + 4} - \tau_k)$, where $\beta_0 := L_f^g$ and $\tau_0 := (\sqrt{5} - 1)/2$. The following corollary illustrates the convergence of Algorithm 1 using $(\text{1P2D}_\mu)$; see [28] for the detail proof.

**Corollary 1.** *Let $f \in \mathcal{F}_\mu$ and $\left\{(\bar{\mathbf{x}}^k, \bar{\mathbf{y}}^k)\right\}_{k \geq 0}$ be generated by Algorithm 1 using* $(\text{1P2D}_\mu)$. *Then:*

$$\|\mathbf{A}\bar{\mathbf{x}}^k - \mathbf{b}\|_2 \leq \frac{4\|\mathbf{A}\|^2}{\mu_f(k+2)^2}D_{\mathcal{Y}}^\star, \ and \ -D_{\mathcal{Y}}^\star\|\mathbf{A}\bar{\mathbf{x}}^k - \mathbf{b}\| \leq f(\bar{\mathbf{x}}^k) - f^\star \leq 0.$$

*Moreover, we also have* $\|\bar{\mathbf{x}}^k - \mathbf{x}^\star\| \leq \frac{4\|\mathbf{A}\|}{(k+2)\mu_f}D_{\mathcal{Y}}^\star$.

It is important to note that, when $f \in \mathcal{F}_\mu$, we only have one smoothness parameter $\beta$ and, hence, we do not need to fix the number of iterations a priori (compared with [18]).

## 4 Algorithmic enhancements through existing methods

Our framework can directly instantiate concrete variants of some popular primal-dual methods for (1). We illustrate three connections here and establish one convergence result for the second variant. We also borrow adaptation heuristics from other algorithms to enhance our practical performance.

**4.1. Proximal-point methods.** We can choose $\mathbf{x}_c^k := \mathbf{x}_{\gamma_{k-1}}^\star(\hat{\mathbf{y}}^{k-1})$. This makes Algorithm 1 similar to the proximal-based decomposition algorithm in [29], which employs the proximal term $d_b(\cdot, \hat{\mathbf{x}}_{k-1}^\star)$ with the Bregman distance $d_b$. The convergence analysis can be found in [28].

**4.2. Primal-dual hybrid gradient (PDHG).** When $f$ is 2-decomposable, i.e., $f(\mathbf{x}) := f_1(\mathbf{x}_1) + f_2(\mathbf{x}_2)$, we can choose $\mathbf{x}_c^k$ by applying one gradient step to the augmented Lagrangian term as:

$$\mathbf{x}_c^k := [\mathbf{g}_1^k; \mathbf{g}_2^k] \text{ with } \begin{cases} \mathbf{g}_1^k & := \mathbf{x}_1^k - \|\mathbf{A}_1\|^{-2}\mathbf{A}_1^T(\mathbf{A}_1\mathbf{x}_1^k + \mathbf{A}_2\mathbf{x}_2^k - \mathbf{b}), \\ \mathbf{g}_2^k & := \mathbf{x}_2^k - \|\mathbf{A}_2\|^{-2}\mathbf{A}_2^T(\mathbf{A}_1\mathbf{x}_1^{k+1} + \mathbf{A}_2\mathbf{x}_2^k - \mathbf{b}). \end{cases} \quad \text{(PADMM)}$$

In this case, (1P2D) leads to a new variant of PADMM in [8] or PDHG in [9].

**Corollary 2** ([28]). *Let $\left\{(\bar{\mathbf{x}}^k, \bar{\mathbf{y}}^k)\right\}_{k \geq 0}$ be a sequence generated by* (1P2D) *in Algorithm 1 using* $\mathbf{x}_c^k$ *as in* (PADMM). *If $\gamma_0 := \frac{2\sqrt{2}\|\mathbf{A}\|_2}{K+1}$ and $c_k := 0$ for all $k = 0, 1, \ldots, K$, then we have*

$$\begin{cases} \|\mathbf{A}\bar{\mathbf{x}}^K - \mathbf{b}\| & \leq \frac{2\sqrt{2}\|\mathbf{A}\|(D_{\mathcal{Y}}^\star + D_{\mathcal{X}})}{(K+1)}, \\ -D_{\mathcal{Y}}^\star\|\mathbf{A}\bar{\mathbf{x}}^K - \mathbf{b}\| \leq f(\bar{\mathbf{x}}^K) - f^\star & \leq \frac{2\sqrt{2}\|\mathbf{A}\|}{(K+1)}D_{\mathcal{X}}^2, \end{cases} \quad (21)$$

*where $D_{\mathcal{X}} := 4\max\{\|\mathbf{x} - \hat{\mathbf{x}}\| : \mathbf{x}, \hat{\mathbf{x}} \in \mathcal{X}\}$.*

**4.4. ADMM.** When $f$ is 2-decomposable as $f(\mathbf{x}) := f_1(\mathbf{x}_1) + f_2(\mathbf{x}_2)$, we can choose $d_b$, $\mathbf{S}$ and $\mathbf{x}_c^k$ such that $d_b(\mathbf{Sx}, \mathbf{Sx}_c) := (1/2)\left[\|\mathbf{A}_1\mathbf{x}_1 + \mathbf{A}_2\mathbf{x}^k - \mathbf{b}\|^2 + \|\mathbf{A}_1\mathbf{x}_1^{k+1} + \mathbf{A}_2\mathbf{x}_2 - \mathbf{b}\|^2\right]$. Then Algorithm 1 reduces to a new variant of ADMM. Its convergence guarantee is fundamentally as same as Corollary 2. More details of the algorithm and its convergence can be found in [28].

**4.5. Enhancements of our schemes.** For the PADMM and ADMM methods, a great deal of adaptation techniques has been proposed to enhance their convergence. We can view some of these techniques in the light of model-based excessive gap condition. For instance, Algorithm 1 decreases the smoothed gap function $G_{\gamma_k\beta_k}$ as illustrated in Definition 1. The actual decrease is then given by $f(\bar{\mathbf{x}}^k) - f^\star \leq \gamma_k(D_{\mathcal{X}}^{\mathbf{S}} - \Psi_k/\gamma_k)$. In practice, $D_k := D_{\mathcal{X}}^{\mathbf{S}} - \Psi_k/\gamma_k$ can be dramatically smaller than $D_{\mathcal{X}}^{\mathbf{S}}$ in the early iterations. This implies that increasing $\gamma_k$ can improve practical performance. Such a strategy indeed forms the basis of many adaptation techniques in PADMM, and ADMM.

Specifically, if $\gamma_k$ increases, then $\tau_k$ also increases and $\beta_k$ decreases. Since $\beta_k$ measures the primal feasibility gap $\mathcal{F}_k := \|\mathbf{A}\bar{\mathbf{x}}^k - \mathbf{b}\|$ due to Lemma 1, we should only increase $\gamma_k$ if the feasibility gap $\mathcal{F}_k$ is relatively high. Indeed, in the case $\mathbf{x}_c^k := [\mathbf{g}_1^k; \mathbf{g}_2^k]$, we can compute the dual feasibility gap as $\mathcal{H}_k := \gamma_k\|\mathbf{A}_1^T\mathbf{A}_2((\hat{\mathbf{x}}_2^\star)_{k+1} - (\hat{\mathbf{x}}_2^\star)_k)\|$. Then, if $\mathcal{F}_k \geq s\mathcal{H}_k$ for some $s > 0$, we increase $\gamma_{k+1} := c\gamma_k$ for some $c > 1$. We use $c_k = c := 1.05$ in practice. We can also decrease the parameter $\gamma_k$ in (1P2D) by $\gamma_{k+1} := (1 - c_k\tau_k)\gamma_k$, where $c_k := d_b(\mathbf{Sx}_{\gamma_k}^\star(\hat{\mathbf{y}}^k), \mathbf{Sx}_c)/D_{\mathcal{X}}^{\mathbf{S}} \in [0, 1]$ after or during the update of $(\bar{\mathbf{x}}^{k+1}, \bar{\mathbf{y}}^{k+1})$ as in (2P1D) if we know the estimate $D_{\mathcal{X}}^{\mathbf{S}}$.

## 5 Numerical illustrations

**5.1. Theoretical vs. practical bounds.** We demonstrate the empirical performance of Algorithm 1 w.r.t. its theoretical bounds via a basic non-overlapping sparse-group basis pursuit problem:

$$\min_{\mathbf{x} \in \mathbb{R}^n}\left\{\sum_{i=1}^{n_g} w_i\|\mathbf{x}_{g_i}\|_2 : \mathbf{Ax} = \mathbf{b}, \|\mathbf{x}\|_\infty \leq \rho\right\}, \quad (22)$$

where $\rho > 0$ is the signal magnitude, and $g_i$ and $w_i$'s are the group indices and weights, respectively.

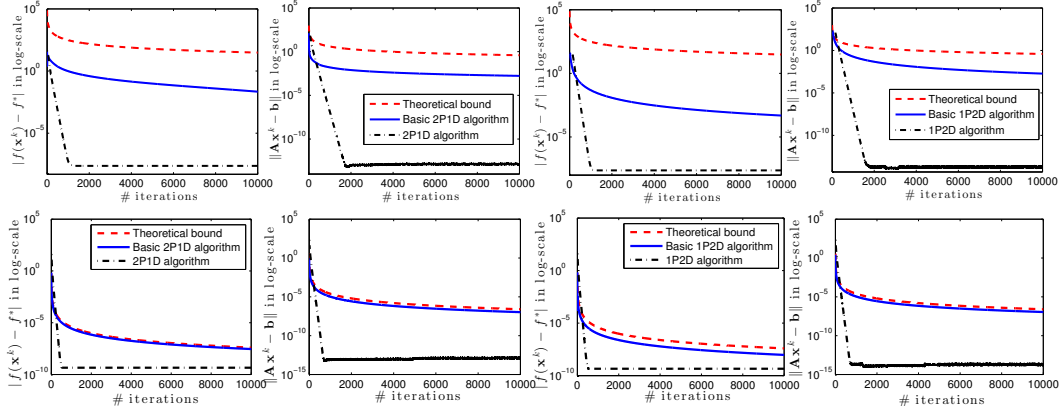

Figure 1: Actual performance vs. theoretical bounds: [top row] the *decomposable* Bregman distance smoother ($\mathbf{S} = \mathbb{I}$) and [bottom row] the augmented Lagrangian smoother ($\mathbf{S} = \mathbf{A}$).

In this test, we fix $\mathbf{x}^c = \mathbf{0}^n$ and $d_b(\mathbf{x}, \mathbf{x}_c) := (1/2)\|\mathbf{x}\|^2$. Since $\rho$ is given, we can evaluate $D_{\mathcal{X}}$ numerically. By solving (22) with the SDPT3 interior-point solver [30] up to the accuracy $10^{-8}$, we can estimate $D_{\mathcal{Y}}^\star$ and $f^\star$. In the (2P1D) scheme, we set $\gamma_0 = \beta_0 = \sqrt{\bar{L}_g}$, while, in the (1P2D) scheme, we set $\gamma_0 := 2\sqrt{2}\|\mathbf{A}\|(K+1)^{-1}$ with $K := 10^4$ and generate the theoretical bounds defined in Theorem 1.

We test the performance of the four variants using a synthetic data: $n = 1024$, $m = \lfloor n/3 \rfloor = 341$, $n_g = \lfloor n/8 \rfloor = 128$, and $\mathbf{x}^\natural$ is a $\lfloor n_g/8 \rfloor$-sparse vector. Matrix $\mathbf{A}$ are generated randomly using the iid standard Gaussian and $\mathbf{b} := \mathbf{A}\mathbf{x}^\natural$. The group indices $g_i$ is also generated randomly ($i = 1, \cdots, n_g$).

The empirical performance of two variants: (2P1D) and (1P2D) of Algorithm 1 is shown in Figure 1. The basic algorithm refers to the case when $\mathbf{x}_c^k := \mathbf{x}_c = 0^n$ and the parameters are not tuned. Hence, the iterations of the basic (1P2D) use only 1 proximal calculation and applies $\mathbf{A}$ and $\mathbf{A}^T$ once each, and the iterations of the basic (2P1D) use 2 proximal calculations and applies $\mathbf{A}$ twice and $\mathbf{A}^T$ once. In contrast, (2P1D) and (1P2D) variants whose iterations require one more application of $\mathbf{A}^T$ for adaptive parameter updates.

As can be seen from Figure 1 (row 1) that the empirical performance of the basic variants roughly follows the $\mathcal{O}(1/k)$ convergence rate in terms of $|f(\bar{\mathbf{x}}^k) - f^\star|$ and $\|\mathbf{A}\bar{\mathbf{x}}^k - \mathbf{b}\|_2$. The deviations from the bound are due to the increasing sparsity of the iterates, which improves empirical convergence. With a kick-factor of $c_k = -0.02/\tau_k$ and adaptive $\mathbf{x}_c^k$, both turned variants (2P1D) and (1P2D) significantly outperform theoretical predictions. Indeed, they approach $\mathbf{x}^\star$ up to $10^{-13}$ accuracy, i.e., $\|\bar{\mathbf{x}}^k - \mathbf{x}^\star\| \leq 10^{-13}$ after a few hundreds of iterations.

Similarly, Figure 1 (row 2) illustrates the actual performance vs. the theoretical bounds $\mathcal{O}(1/k^2)$ by using the augmented Lagrangian smoother. Here, we solve the subproblems (13) and (17) by using FISTA [31] up to $10^{-8}$ accuracy as suggested in [28]. In this case, the theoretical bounds and the actual performance of the basis variants are very close to each other both in terms of $|f(\bar{\mathbf{x}}^k) - f^\star|$ and $\|\mathbf{A}\bar{\mathbf{x}}^k - \mathbf{b}\|_2$. When the parameter $\gamma_k$ is updated, the algorithms exhibit a better performance.

**5.2. Binary linear support vector machine.**    This example is concerned with the following binary linear support vector machine problem:

$$\min_{\mathbf{x} \in \mathbb{R}^n} \left\{ F(\mathbf{x}) := \sum_{j=1}^m \ell_j(y_j, \mathbf{w}_j^T \mathbf{x} - \mathbf{b}_j) + g(\mathbf{x}) \right\}, \tag{23}$$

where $\ell_j(s, \tau)$ is the Hinge loss function given by $\ell_j(s, \tau) := \max\{0, 1 - s\tau\} = [1 - s\tau]_+$, $\mathbf{w}_j$ is the column of a given matrix $\mathbf{W} \in \mathbb{R}^{m \times n}$, $\mathbf{b} \in \mathbb{R}^n$ is the bias vector, $\mathbf{y} \in \{-1, +1\}^m$ is a classifier vector $g$ is a given regularization function, e.g., $g(\mathbf{x}) := (\lambda/2)\|\mathbf{x}\|^2$ for the $\ell_2$-regularizer or $g(\mathbf{x}) := \lambda\|\mathbf{x}\|_1$ for the $\ell_1$-regularizer, where $\lambda > 0$ is a regularization parameter.

By introducing a slack variable $\mathbf{r} = \mathbf{W}\mathbf{x} - \mathbf{b}$, we can write (23) in terms of (1) as:

$$\min_{\mathbf{x} \in \mathbb{R}^n, \mathbf{r} \in \mathbb{R}^m} \left\{ \sum_{j=1}^m \ell_j(y_j, \mathbf{r}_j) + g(\mathbf{x}) : \mathbf{W}\mathbf{x} - \mathbf{r} = \mathbf{b} \right\}. \tag{24}$$

Now, we apply the (1P2D) variant to solve (24). We test this algorithm on (24) and compare it with LibSVM [32] using two problems from the LibSVM data set available at http://www.csie. ntu.edu.tw/~cjlin/libsvmtools/datasets/. The first problem is a1a, which has $p = 119$ features and $N = 1605$ data points, while the second problem is news20, which has $p = 1'355'191$ features and $N = 19'996$ data points.

We compare Algorithm 1 and the LibSVM solver in terms of the final value $F(\mathbf{x}^k)$ of the original objective function $F$, the computational time, and the classification accuracy $ca_\lambda := 1 - N^{-1} \sum_{j=1}^{N} \left[ \text{sign}(\mathbf{W}\mathbf{x}^k - \mathbf{r}) \neq \mathbf{y} \right]$ of both training and test data set. We randomly select $30\%$ data in a1a and news20 to form a test set, and the remaining $70\%$ data is used for training. We perform 10 runs and compute the average results. These average results are plotted in Fig. 2 for two separate problems, respectively. The upper and lower bounds show the maximum and minimum values of these 10 runs.

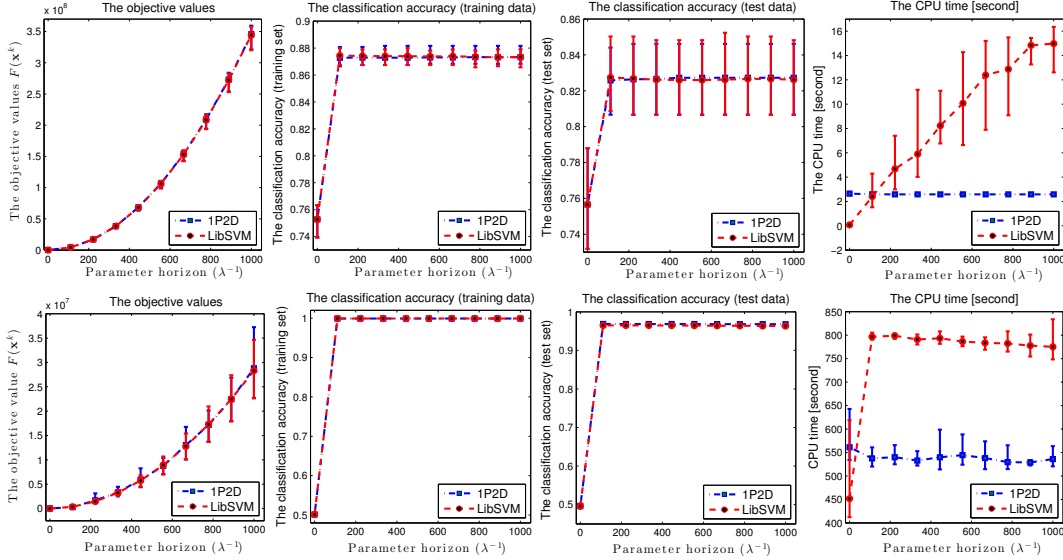

Figure 2: The average performance results of the two algorithms on the a1a (first row) and news20 (second row) problems.

As can be seen from these results that both solvers give relatively the same objective values, the accuracy for these two problems, while the computational of (1P2D) is much lower than LibSVM. We note that LibSVM becomes slower when the parameter $\lambda$ becomes smaller due to its active-set strategy. The (1P2D) algorithm is almost independent of the regularization parameter $\lambda$, which is different from active-set methods. In addition, the performance of (1P2D) can be improved by taking account its parallelization ability, which has not fully been exploited yet in our implementation.

## 6 Conclusions

We propose a model-based excessive gap (MEG) technique for constructing and analyzing first-order primal-dual methods that numerically approximate an optimal solution of constrained convex optimization problems (1). Thanks to a combination of smoothing strategies and MEG, we propose, to the best of our knowledge, the first primal-dual algorithmic schemes for (1) that theoretically obtain optimal convergence rates directly without averaging the iterates and that seamlessly handle the $p$-decomposability structure. In addition, our analysis techniques can be simply adapt to handle inexact oracle produced by solving approximately the primal subproblems (c.f. [28]), which is important for the augmented Lagrangian versions with lower-iteration counts. We expect a deeper understanding of MEG and different smoothing strategies to help us in tailoring adaptive update strategies for our schemes (as well as several other connected and well-known schemes) in order to further improve the empirical performance.

**Acknowledgments.** This work is supported in part by the European Commission under the grants MIRG-268398 and ERC Future Proof, and by the Swiss Science Foundation under the grants SNF 200021-132548, SNF 200021-146750 and SNF CRSII2-147633.

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
