[Reviews · NeurIPS 2014]

Submitted by Assigned_Reviewer_32

The idea of this paper is to apply Nesterov's excessive gap technique to linearly constrained convex optimization problems which occur in many applications. This idea seems novel and useful since it allows the authors to prove convergence rates, recover existing methods, and derive new algorithms. The mathematical quality of the paper is high, many interesting special cases and connections to existing methods are discovered and proofs are provided in the supplementary material.

The paper feels a bit dense at times and some things could be explained in a clearer way. This is important to make the paper more accessible for a wider NIPS audience and to allow readers to quickly implement and compare the new algorithms proposed here. Sections 2.3 and 3.2 are quite hard to read. As an example, it would be important to provide a more self-contained motivation of the excessive gap technique in 3.2. Another example is that until Section 4, it is not mentioned that the choice of x_c is important and how one should choose it. For a practitioner who looks at Algorithm 1 first this might be a problem. The algorithmic framework proposed here is quite general and thus has quite a lot parameters and different variables. Unfortunately, they are often quite hard to find in the text, e.g., the important mappings x^*_{\gamma}(\bar{y}) and x^{\delta}_{\gamma}(\bar{y}) should be featured more prominently since the are now a bit hidden in the text on p. 4.

The authors provide a wide range of numerical experiments. Concerning the numerical experiments provided in the appendix, I am a bit puzzled by the failure of ADMM on the TV-regularized deblurring problem. It is guaranteed to converge so I am wondering if it is that slow or if there is a problem in the code. The remark on line 1114 indicates the latter since it should converge for all step length parameters (>0).
Another issue in this context is the comparison of ADMM, PADMM and the corresponding new algorithms proposed in Figure 4. It is clear that the strength of PADMM vs. ADMM cannot be seen in the number of iterations since ADMM solves a linear system and PADMM not. Of course, this is especially important if one cannot use a fast algorithm like the FFT but even in this special case plotting w.r.t. computation time might be more suitable.
For the basis pursuit problem the algorithms the authors compared to are not the fastest any more. Algorithms which combine gradient-descent and prox-operations with active set techniques, e.g., often perform much better.
In general, it might have been better to focus on a couple of experiments and treat them in more detail, e.g., explicitly comparing (81) with ADMM to show the differences. If the new algorithms perform as good or better as very similar existing methods, one should give the reader a clear understanding what the differences and advantages are.

The authors might also want to have a look at these two papers and mention the relations to their work if necessary:

"Combining Lagrangian Decomposition and Excessive Gap Smoothing Technique for Solving Large-Scale Separable Convex Optimization Problems" by Dinh et al.

"Diagonal preconditioning for first order primal-dual algorithms in convex optimization" by T. Pock and A. Chambolle

Typo: l. 235: Step 4 instead of Step 5
Summary: This paper is of high quality and contains novel ideas. The only concern I have is that it is not easily accessible to a larger audience. Together with the very long appendix it seems more like a math journal paper and lacks a bit in motivating the new techniques and in pointing out the differences to existing work.

Submitted by Assigned_Reviewer_36

=============added after rebuttal===========
The authors are correct; I was mistaken about the correctness of Lemma 3. My confusion arose from line 528, where the authors denoted by x_gamma^* as THE solution of eq. (23). Perhaps it is worth pointing out that the solution of eq. (23) need not be unique, although they all have the same product Ax_gamma^* = const, from which Lemma 3 would be easy to follow.
====================================

This work extended Nesterov's excessive gap technique to handle additional linear constraints. Unlike most existing methods, the convergence rates in terms of both objective value and linear constraint satisfiability are established, matching the best of the known ones. The proposed algorithm can naturally exploit the decomposability of the nonsmooth objective function. Experiments demonstrate that the proposed algorithms compare favorably against some existing ones.

The reviewer found the idea to bound the primal optimality gap and linear constraint satisfiability separately interesting and useful. The proved rates also seem to match the best known ones. However, the presentation of this paper needs some serious improvement. The authors put a lot of stuff in the main paper, but due to the lack of space, very few are explained in details. It would perhaps make more sense to cut some results out and explain the key things more clearly. For instance, the similarity and difference with existing ADMM approaches worth a detailed elaboration.

It is quite annoying that almost all references in the paper are misplaced: e.g., line 107, pretty sure [3] has nothing to do with smoothing; line 109, [6] has nothing to do with ADMM; line 118, [3] is meant to be [1], etc. This cross-reference issue definitely needs to be fixed. Also, the proximal tractability assumption in line 049 refers to specifically the Euclidean norm, but later on the paper focuses on a general strongly convex function b(.). Some consistency would be appreciated. It should be possible to absorb the constraint X into the objective function f, i.e., f <--- f + indicator(X). Conceptually this is simpler.

Lemma 3 in the appendix does not seem to be correct: Simply let A=0, b=0, g_gamma then can certainly be nonsmooth. One needs to assume A to have linearly independent columns, which unfortunately may not hold in applications. This put some doubt on the correctness of the first part of Theorem 1. Suppose one can fix this issue by assuming A has full column rank, it should be pointed out that the seemingly "better" rate 1/k^2 in Theorem 1 is actually quite useless since it requires solving some hard proximal problem which usually can only be solved using an iterative procedure (such as FISTA used in the experiment). One should also caution against the proved rates since some of them appear to contradict some lower bounds. For instance the 1/k^2 rate in Corollary 1 is impossible had the iterates been satisfying the linear constraints.

The experiments in the main paper are insufficient. Comparison against state-of-the-art alternatives (some included in the appendix) needs to be included. To save space, the strongly convex experiment perhaps can go to the appendix instead, as the improvement appear to be marginal.
Summary: This work extended Nesterov's excessive gap method to handle additional linear constraints. Through smoothing the O(1/t) convergence rate is proved for both the objective value and the linear constraint satisfiability. The results are of some interest, but the presentation can be much improved.

Submitted by Assigned_Reviewer_41

The paper proposes a primal dual optimization scheme for decomposable problems with linear coupling constraints, which can recover augmented Lagrangian, ADMM and fast dual descent methods.
The paper is novel and highly non-trivial, but incremental in the sense that it provides convergence rates for the iterates themselves, avoiding the need to average as in existing methods.
Unfortunately, the paper is hard to read due to the many technical details in the main part. Meaningful experiments are provided in the supplementary material only.
The presented theory only applies with known accuracy for the subproblem solutions in each iteration, however the experiments (e.g. 1P2D) only use a few inexact steps.

* The supplementary material contains a large collection of examples of applications, and these experiments seem encouraging for the presented method. Still, for all applications I do miss a comparison to the naive approach of just penalizing ||Ax-b||^2 in the objective and running standard accelerated gradient methods. Also, the experiments do not compare to other amenable methods, only standard ADMM in some cases.
Instead of including just one experiment of the 1P2D method in the main paper, compared to the theoretical bound, it would be much more valuable to have some practical experiments comparing to competing methods, such as e.g. the SVM or Robust PCA case as presented in the appendix currently.

* Another critique point I have is that the iteration complexity of the proposed methods are not discussed in the paper. While the theory only holds for known accuracy for the sub-problems / prox-operators, the experiments are performed with arbitrary inexact subproblems (12) or (15) or prox operators, e.g. a few FISTA iterations, which is not covered by the presented theory.
(Update after author feedback: Unfortunately the feedback has not precisely addressed the computational complexity or the accuracy level of the subproblems. Hope this will be made more precise in the paper in case of acceptance.)

* Structural assumption of decomposability:
It is important to motivate the considered class of optimization problems.
The assumption (3) alone is strong as it requires both sum structure as well as dependence of each summands on only one corresponding coordinate block. Using auxiliary variables, many ML problems can be transformed to form (1), which is however only explained in the lengthy supplementary material. Please add in the text that this is indeed possible for applications of interest. Similarly: When mentioning "we employ (3) to do away with the X -feasibility of the algorithmic iterates and focus on linear constraints": add a comment which important applications then still fall into the setting. The relation to Nesterov [1] could be made more clear also. Furthermore, it should be stated early that standard gradient methods do not apply to problems of the form (1) (even assuming prox tractability) due to the linear coupling constraints. However, they do apply for the penalized versions when just adding ||Ax-b|| to the objective.

The abstract is not written precisely as to reveal the problem setting. You should clarify that your method can deal with linear coupling constraints, but assumes separable objective.

* Introduction paragraph on comparison to [3]: This seems the wrong reference, I see no connection of the concepts used there to optimization setting here.

* Adaptive proximal-centers are not explained, and it's not clear if the presented theory applies to this case as well.

* Add references when mentioning "immediately supports parallel and distributed implementations in synchronous hardware architectures", as this is non-trivial.

### Minor details:

-l43: you might want to add that x_i \in X_i, or that the parts of coordinates x_i of x form a partition. And maybe use cartesian product notation instead of pi product for X.

-l106: slightly tune down "radically different"

-l118: +more structured

-l201: phi is not defined here when introducing Nesterov's setting

-ref [17]: venue missing
Summary: The paper proposes a primal dual optimization scheme for decomposable problems with linear coupling constraints, which can recover augmented Lagrangian, ADMM and fast dual descent methods.
The paper is novel and highly non-trivial, but incremental in the sense that it provides convergence rates for the iterates themselves, avoiding the need to average as in existing methods. The presented theory only applies with known accuracy for the subproblem solutions in each iteration, however the experiments (e.g. 1P2D) only use a few inexact steps.
Author Feedback
Author rebuttal: We would like to thank all the reviewers for their constructive and encouraging comments. We agree that the paper is rather mathematical but there is enough guidance in all three reviews to improve the presentation and motivation to make it accessible to a broader audience, fix the typos and reference issues.

R#1:
1/ ADMM on the TV-regularized deblurring problem…

The ADMM code is directly from the reference [7]. The code has an aggressive way of updating the penalty parameter, which can lead to instability. We use this example to highlight the fact that the regularization parameter choice is more like an art and science. When it works, it is great. However, it prevents the ADMM like methods from becoming reliable black-box methods that we can further build upon.

2/ … the comparison of ADMM, PADMM…

Iteration comparison is only meaningful when per-iteration times are similar, which is indeed the case for this. Fortunately, we have the timings and can easily add them for the appendix.

3/ …BP & active sets…

Please let us know a reference. We would be happy to compare them for the BP example in the appendix.

4/ … better to focus on a couple of experiments…

We have thought about this very carefully when making the submission. We believe our main contribution is theoretical and we chose to illustrate the bounds.

In fact, we find it extremely hard to make a fair comparison to existing methods by presenting one or two particular examples without tuning of the algorithms that exploit application knowledge. This by itself requires additional space, which we could not afford.

R#2:
1/ Lemma 3 in the appendix does not seem to be correct…

We think that the reviewer is mistaken. If A=0 and b=0, then g_gamma does not depend on y, and becomes a constant function. In such a case, g_gamma is of course still concave and smooth. Its gradient is zero, which is certainly Lipschitz continuous. In other cases, we require that {Ax = b} cap X is nonempty. The proof of these properties of g_gamma can be found in, e.g., the references [2] (appendix), [26] (main paper) and G. Lan & R. Monteiro (2013).
Corollary 1 proves the optimal rate of the strongly convex case, which is similar to optimal gradient methods but for both objective residual and feasibility gap.

2/ Comparison against state-of-the-art alternatives …

Please see our remark to R#1.

R#3:

1/ known accuracy for the subproblem solutions in each iteration, however the experiments (e.g. 1P2D) only use a few inexact steps.

The analysis shows that our method is numerically stable to the inexactness of the subproblem solver. Interestingly, we noticed that the subproblem solver produces a high-accuracy solution thanks to warm-start strategy. Otherwise, we simply need to use more iterations by using the primal domain radius.

2/ …the naive approach of just penalizing ||Ax-b||^2…

Along with this experiment, we must also justify the choice of the regularization parameter. This is a very deep topic, which distracts from our theoretical results. We indeed compared against many methods such as TFOCS (which implements FISTA with adaptive restart and linesearch), SPGL1, L1General and Sparsa, which are available online. However, such methods are not really successful as YALL1 (see the reference [9] in appendix for detail comparisons). Hence, we take the best-known code, YALL1 for our comparison.

3/ The abstract is not written precisely as to reveal the problem setting...

We will change it according to the reviewer’s comment.

4/ Adaptive proximal-centers are not explained, and it's not clear if the presented theory applies to this case as well.

Our theory indeed does apply. As an example, see the theory for the PDHG variant, which uses an adaptive center point by taking gradient step of the quadratic term.